# Biomarkers of Dementia with Lewy Bodies: Differential Diagnostic with Alzheimer’s Disease

**DOI:** 10.3390/ijms23126371

**Published:** 2022-06-07

**Authors:** Olivier Bousiges, Frédéric Blanc

**Affiliations:** 1Laboratory of Biochemistry and Molecular Biology, University Hospital of Strasbourg, 67000 Strasbourg, France; 2Team IMIS, ICube Laboratory UMR 7357 and FMTS (Fédération de Médecine Translationnelle de Strasbourg), University of Strasbourg and CNRS, 67000 Strasbourg, France; f.blanc@unistra.fr; 3CM2R (Research and Resources Memory Centre), Geriatrics Department, Day Hospital and Cognitive-Behavioral Unit University Hospitals of Strasbourg, 67000 Strasbourg, France

**Keywords:** dementia with lewy bodies, Alzheimer’s disease, cerebro-spinal fluid, biomarkers, α-synuclein, brain MRI, FP-CIT SPECT, insula, RT-QuIC

## Abstract

Dementia with Lewy Bodies (DLB) is a common form of cognitive neurodegenerative disease. Only one third of patients are correctly diagnosed due to the clinical similarity mainly with Alzheimer’s disease (AD). In this review, we evaluate the interest of different biomarkers: cerebrospinal fluid (CSF), brain MRI, FP-CIT SPECT, MIBG SPECT, PET by focusing more specifically on differential diagnosis between DLB and AD. FP-CIT SPECT is of high interest to discriminate DLB and AD, but not at the prodromal stage (i.e., MCI). MIBG SPECT with decreased cardiac sympathetic activity, perfusion SPECT with occipital hypoperfusion, FDG PET with occipital hypometabolism and cingulate island signs are of interest at the dementia stage but with a lower validity. Brain MRI has shown differences in group study with lower grey matter concentration of the Insula in prodromal DLB, but its interest in clinical routines is not demonstrated. Concerning CSF biomarkers, many studies have already examined the relevance of AD biomarkers but also alpha-synuclein assays in DLB, so we will focus as comprehensively as possible on other biomarkers (especially those that do not appear to be directly related to synucleinopathy) that may be of interest in the differential diagnosis between AD and DLB. Furthermore, we would like to highlight the growing interest in CSF synuclein RT-QuIC, which seems to be an excellent discrimination tool but its application in clinical routine remains to be demonstrated, given the non-automation of the process.

## 1. Introduction

### 1.1. Epidemiology of DLB

Dementia with Lewy bodies (DLB) is the main neurodegenerative pathology affecting cognitive functions in the elderly after Alzheimer’s disease (AD) [1]. It begins after the age of 50 and represents 20% of demented patients [2]. First described in 1961 [3], it is only recently that it has been recognized as a common cause of dementia [4].

### 1.2. Diagnostic Criteria

DLB diagnostics criteria have evolved over time and were revised in 2005 and again in 2017 [5,6]. The McKeith criteria for the diagnosis of probable DLB are, in addition to cognitive decline (essential criterion), at least two core criteria and for the possible diagnosis either a core criterion with or without suggestive criteria, or at least one suggestive criterion (Table 1). McKeith and the international DLB consortium have also defined diagnostic criteria for prodromal forms [7]. The prodromal stage of DLB (pro-DLB), also called mild cognitive impairment due to Lewy bodies has been described in detail: the first criteria of this prodromal stage are similar to the stage of dementia with the difference that a decrease in functional capacity is either non-existent or minimal. So pro-DLB can be defined as the presence of the disease but cognitive impairment is not sufficient to lead to functional deficits in activities of daily living, and thus to dementia. Descriptions of pro-DLB criteria have been proposed [8]: pro-DLB can be defined as those patients who meet the revised diagnostic criteria for DLB but instead of dementia, fit the criteria for mild cognitive impairment (MCI) [7].

### 1.3. Differential Diagnosis with AD and PD

Due to some clinical and neuropsychological similarities with AD, the diagnosis of DLB is not straightforward. The main confounding pathologies are AD, vascular dementia, Parkinson’s disease dementia (PDD), frontotemporal lobe degeneration (FTLD), progressive supranuclear palsy (PSP), multisystem atrophy (MSA), corticobasal degeneration (CBD) and prion diseases for brain diseases and recurrent depression, schizophrenia, bipolar disorders for psychiatric diseases. The most common differential diagnosis for the clinician remains with AD.

DLB is close to Parkinson’s disease (PD) due to the presence of Parkinsonism (bradykinesia, rigidity, postural instability and resting tremor), which is often discrete, especially at the beginning of the disease [7,8]. DLB is also close to PD because of its pathophysiology, with the presence of positive α-synuclein (α-syn) aggregates in the brain, forming Lewy bodies [9]. Thus, PD and DLB, due to common aggregation processes, form a set of pathologies called synucleinopathies, including also PDD and multiple system atrophy (MSA). The pathophysiological difference that can be noted between DLB and PD is the aggregates localization. While α-syn aggregates are mostly localized in the brainstem and in the substantia nigra at the beginning of PD, they are rather diffuse throughout the brain in the early stages of DLB. Moreover, the three-dimensional structure of α-syn is different from the two others synucleinopathies: PD and MSA [10]. The differential diagnosis between PDD and DLB depends on the timing of events between the cognitive and motor impairment. In PDD, the cognitive symptoms appear after the parkinsonian syndrome, whereas in DLB the motor deficits appear after the cognitive deficits. Thus, according to an arbitrary rule, if the cognitive deficits appear at least one year after the motor deficits it will be PDD, otherwise it will be DLB [4,6].

Many symptoms of DLB are close to those of AD, especially at the onset of the pathology: executive functions, visual memory, visuo-constructive and visuospatial abilities with weaknesses for episodic memory, short-term and working memory, verbal initiation, praxis, language, as well as social cognition [11]. DLB and AD can often be associated: 87% of DLB patients have moderate to abundant cortical amyloid plaques [12]. Thus, in these patients with AD/DLB diseases, DLB being masked, this disease is more difficult to diagnose.

DLB presents certain clinical specificities such as visual illusions or hallucinations and fluctuations in attention, but also a particular sensitivity to neuroleptics [5,8]. Despite the very high diagnostic specificity of these criteria (the specificity of probable DLB is 95.1% in early stages and 88% in late stages) [4,13], their sensitivity remains low (32%) in pure DLB or even lower (12%) when associated with AD [14]. In other words, DLB is still a largely underdiagnosed disease in more than two-thirds of cases. It is therefore essential to discover new biomarkers that can distinguish DLB from AD to improve differential diagnosis.

### 1.4. Therapeutic Management

The treatment of DLB patients consists of a combination of pharmacological and non-pharmacological treatments. At the pharmacological level, each symptom has its own symptomatic treatment: donepezil or rivastigmine for cognitive disorders and cognitive fluctuations, levodopa in small doses for Parkinsonism, melatonin for behavioral disorders in REM sleep behavior disorders (RBD), clozapine, quetiapine and pimavanserine for invasive hallucinations or delusion [15]. Other antipsychotics are contraindicated because they are poorly tolerated: There is a risk of increased cognitive impairment, falls, increased parkinsonian syndrome, confusion, neuroleptic malignant syndrome and even death [16]. On a non-pharmacological level, cognitive remediation is used to combat cognitive disorders, and physical therapy to combat motor disorders and falls.

### 1.5. Diagnostic Tools Available at the Present Time

In this review, we will focus on both imaging (MRI, scintigraphy) and CSF biomarkers. We will see that damage to the insular cortex and preservation of the hippocampus is rather in favor of a DLB. An abnormal FP-CIT SPECT or MIBG appears to be in favor of a DLB and so seems to be quite good (at a late stage) for differentiating DLB from AD. Concerning CSF, many publications have focused on the study of the AD biomarkers (Aβ42, t-Tau, phospho-Tau181, and Aβ40) in dementias other than AD, and in particular in DLB patients. We have previously reviewed the relevance of these biomarker assays in the differential diagnosis between AD and DLB [17]. Therefore, to summarize, the CSF biomarkers used for AD diagnosis are the only ones available in hospital routine and their usefulness is undeniable in the differential diagnosis with DLB, t-Tau and phospho-Tau181 being the most discriminating biomarkers between these two pathologies. Furthermore, an Aβ42 pathological level (without any other pathological biomarker) is in favor of DLB diagnosis. Furthermore, in patients with DLB, a low value of Aβ42 is in favor of a relatively advanced stage of the disease, since in the prodromal stage, the patient’s CSF shows only a little or no Aβ42 decrease. Of note, half of DLB patients have no pathological biomarkers at the prodromal stage, and more than a quarter at the demented stage [18]. We also noted that the Aβ42/Aβ40 ratio is an excellent discrimination tool between AD and DLB, especially at the prodromal stage. In addition, the total α-synuclein seemed to be a good biomarker because of its role in the DLB pathophysiology. Studies show that α-synuclein may actually be relevant in discriminating AD from DLB, but the changes do not come from DLB patients (not significantly different from controls), but from AD patients with higher levels of α-synuclein in the CSF of AD patients compared to DLB and control patients [19,20,21]. Thus, it seems that the α-syn assay in CSF would rather strengthen the AD diagnosis. On the other hand, α-syn still does not clearly distinguish DLB from other demented pathologies or from control patients.

However, there are other biomarkers of high interest for the diagnosis of DLB. First, the slowing of the dominant EEG rhythm assessed visually or with a quantitative approach can differentiate DLB from AD with a sensitivity of 90% and a specificity of 90%. The EEG is therefore a test of choice in the diagnostic toolbox for DLB [22]. Second, when polysomnography shows REM sleep without atonia in a patient with dementia and clinically diagnosed RBD, there is a 90% likelihood of synucleinopathy [6].

However, despite these interesting results in the differential diagnosis of DLB, these biomarkers are not specific enough for DLB and the search for new biomarkers for this disease is of great importance. Thus, this review will look at all biomarkers other than Alzheimer’s biomarkers (Aβ42, Aβ40, T-Tau, P-Tau) or the total α-syn assay or the α-syn post-translational modifications assay. Other biomarkers, although not necessarily specific to DLB or AD, seem interesting for discriminating these two pathologies. We will discuss the relevance of these new biomarkers. We will also look at a promising technique, namely α-syn RT-QuIC.

## 2. Methodology

The bibliographic search was conducted on Pubmed. Different keywords were typed for this search, such as “biomarker”, “Cerebrospinal fluid”, “CSF”, “Imaging”, “SPECT”(also: “single-photon-emission computed tomography”), “MIBG”, “PET”, “MRI”, “RT-QuIC” AND “Dementia with Lewy Bodies” (in the title) AND “Alzheimer’s disease” (in the title and abstract). The results of this research were between 1950 and 14 February 2022 (date of the last extraction on pubmed).

We use the terms sensitivity and specificity many times in this review. As a reminder, the sensitivity of a test is the ability of that test to correctly diagnose ill patients, while specificity is the ability of a test to correctly reject healthy patients and so to do a correct diagnosis.

## 3. Brain Imaging Diagnostic Biomarkers

### 3.1. Brain MRI

The prodromal stage of AD in the brain MRI group study shows greater hippocampal and parietal atrophy [23]. In addition, insular atrophy seems to be an interesting marker of prodromal DLB when compared to prodromal AD and healthy elderly controls (Figure 1) [23,24,25]. However, analysis with voxel-based morphometry (VBM) and cortical thickness on the groups of patients is not the same as visual and individual analysis. A recent study did not demonstrate differences in the insula and hippocampi between prodromal DLB and AD patients [26]. In this previous study, amyloid and tau biomarkers (CSF or PET) were not done for the two groups. However, when considering the differential diagnosis between prodromal DLB and prodromal AD, preserved hippocampal volumes are associated with an increased risk of probable DLB [27]. Moreover, at the ADPD congress, we showed our work on the visual analysis of AD and DLB prodromal patients: our first results show that in DLB there is isolated atrophy of the anterosuperior part of the insulae, while in AD there is hippocampal and insular atrophy [28].

The dementia stage of AD on brain MRI also shows greater hippocampal atrophy, with the possibility to differentiate AD from DLB with very good sensitivity (91%) and specificity (94%) [29]. However, this is valid for patients near death in moderately severe to severe stages (MMSE<15/30). At earlier stages (i.e., mild to moderate dementia with MMSE ≥ 15/30), hippocampal atrophy is less informative: a sensitivity of 64% and a specificity of 68% [30]. Recent studies have shown an interest to look into the loss of the swallow tail sign. SWI brain MRI usually detects a posterior hyperintense signal area in the substantia nigra of healthy control: the hyperintense signal in between the hypointense signal of the substantia nigra is named the swallow tail sign. Two studies with 15 [31] and 19 [32] DLB patients demonstrated different results when compared to AD patients. Rizzo et al. have shown a sensitivity of 80% and a specificity of 64% [31], Shams et al., a sensitivity 63% and a specificity of 75% [32]. Other studies are necessary to better know the interest of this loss of the swallow tail sign: the search for this sign can be difficult because a thinning of the substantia nigra can also appear in DLB (Figure 1B).

Thus, the clinician should look for insular atrophy without hippocampal atrophy in mild cognitive impairment (MCI) patients, and if present, the clinician must search for the cardinal symptoms of DLB, fluctuations, hallucinations, RBDs and Parkinsonism, including discrete ones [33]. In the case of suspected DLB at the dementia stage, the degree of atrophy is of little importance, apart from hippocampal atrophy, which is often less than in the rest of the brain.

### 3.2. Scintigraphy

#### 3.2.1. FP-CIT SPECT

Nigral dopaminergic cell loss (and not synuclein) was associated with lower striatal I-2beta-carbomethoxy-3beta-(4-iodophenyl)-N-(3-fluoropropyl) nortropane ((123)I-FP-CIT) uptake [34]. FP-CIT SPECT has an excellent accuracy particularly at the stage of moderate dementia: 88% sensitivity and 100% specificity (neuropathologic analysis) [35]. A phase 3 study to validate FP-CIT SPECT for the diagnosis of DLB has demonstrated that in probable DLB (i.e., two core clinical signs), the sensitivity was 77.7% and the specificity was 90.4% [36]. These patients had a mean MMSE of 20 in favor of mild dementia. Interestingly, for possible DLB (i.e., one core clinical sign), a situation a biomarker would be of high interest and the sensitivity was lower (38.2%). Moreover, an abnormal FP-CIT SPECT does not exclude the diagnosis of FTLD. Thus, when FTLD is the control group, specificity for DLB is only 70%, and sensitivity remains excellent (90%) [37]. Despite the limitations cited, a recent meta-analysis confirms that FP-CIT has clear value as a biomarker of DLB at the stage of dementia [38].

The diagnostic value of FP-CIT SPECT for prodromal DLB is lower. In a study of 48 prodromal DLB, sensitivity was 54.2% for probable and possible prodromal DLB, and specificity was 89% [39]. For probable prodromal DLB, the sensitivity was better (61%). In other words, when the FP-CIT SPECT is abnormal in an MCI patient, it is of interest, but if normal, it is not useful in the diagnostic process.

For the clinician, the visual analysis of the FP-CIT SPECT must be deepened by looking at the different parts of the striatum, the caudate nucleus in the anterior part and the putamen in the posterior part (Figure 2A,B). The FP-CIT SPECT is of interest when it is abnormal, especially at the prodromal stage, but when it is normal, it does not eliminate the diagnosis of DLB.

#### 3.2.2. Perfusion SPECT

Perfusion SPECT is an imaging technique for estimating brain perfusion. The main handicap of this technique is its low sensitivity in the context of DLB. Thus, the differentiation between DLB and AD obtained a sensitivity of 65% according to Lobotesis [40] and Pasquier [41] and 74% according to Hanyu [42], using occipital hypoperfusion as a biomarker of DLB. The specificity was better: 82% according to Hanyu. Using image processing with software SPECT Z-score maps with a focus on the cingulate island sign (CIS, see infra), the differential diagnosis between DLB and AD had a sensitivity and specificity of 92.3, and 76.9% [43]. However, these results were lower in a second more powerful study with sensitivity and specificity of 50 and 73% [44].

For the clinician, the hypoperfusion seen in DLB in the dementia stage is often diffuse and partial of the frontal, temporal, parietal, insular and occipital lobes (Figure 2C). Therefore, the clinician should look specifically for the existence of occipital hypoperfusion.

#### 3.2.3. FDG-PET

As in perfusion SPECT occipital hypometabolism in DLB using FDG-PET is of interest and can help to discriminate DLB from AD [45]. The CIS has been proposed as a neuroimaging feature of DLB. The differences between DLB and AD were described in 1997 [46], and the concept of CIS was developed thereafter. Its definition is the hypometabolism of precuneus and cuneus but with sparing of the posterior cingulate cortex [47]. The CIS was found to be correlated with hippocampal atrophy, suggesting a link with AD pathology [48]. Lim et al. [47] reported that the sensitivities for using CIS to diagnose DLB ranged from 62 to 86% and only 43 to 50% for hypometabolism in the medial occipital lobe. Both (occipital hypometabolism and CIS) were demonstrated to have a better sensitivity of 77% and specificity of 80% [47]. Posterior cortical atrophy, one of the clinical subtypes of AD, and DLB show overlapping patterns of hypometabolism including occipital and posterior cingulate [49].

For the clinician, FDG-PET is better than perfusion SPECT for the diagnosis if the search for occipital hypometabolism and sparing of the posterior cingulate regions is performed, while considering the often diffused and decreased metabolism in PET-FDG.

#### 3.2.4. Synuclein PET

There are many efforts to obtain a specific biomarker for synucleopathies in PET [50]. Alpha-synuclein PET is still missing. Many groups are trying to obtain it because it would be a kind of revolution to obtain such a biomarker. Recently, Kuebler et al. have demonstrated the capacity of MODAG-001 to bind to synuclein fibrils in a rat brain using in vivo PET [51]. Previously, Kikochi et al. had shown that synuclein PET could be used to visualize synuclein deposits in eight MSA patients [52]. More recently, Oskar Hansson presented at the ADPD congress 2022 synuclein PET (ACI 12589) results again in eight MSA patients compared to healthy controls, Parkinson’s disease patients and DLB patients. Only the MSA patients had significant labeling, especially in the cerebellar peduncles.

#### 3.2.5. 123 I-Metaiodobenzylguanidine (MIBG) Myocardial Scintigraphy

MIBG myocardial scintigraphy is an imaging technique for estimating sympathetic nerve damage, which post-mortem studies show as reduced in DLB. These damages are seen in primary heart disease, diabetic neuropathy, and also PD and DLB [42]. For DLB at the stage of dementia, the sensitivity ranges from 68.9% [53] N = 61 DLB (ratio method) to 100% N = 19 DLB (ratio method) [42]. The specificity of MIBG scintigraphy when compared to healthy controls or AD is usually excellent: 87% [53], to 92% [42] N = 19 DLB, ratio method [42,53]. Although MIBG scintigraphy seems to be informative, there is a lack of phase 3 studies to validate this biomarker in its use as a differential diagnostic tool between DLB and AD [54].

As for FP-CIT SPECT, the diagnostic value of MIBG scintigraphy is lower in the context of prodromal DLB. In a study of 52 patients with prodromal DLB, sensitivity was 46.2% for probable and possible prodromal DLB, and specificity of 88% (controls: prodromal AD patients) [55].

Table 2 summarises the neuroimaging diagnosis of DLB patients.

## 4. CSF Biomarkers

CSF biomarkers in relation to AD biomarkers or total alpha-synuclein and its post-translational modifications have been addressed previously [17]. Here, we propose to discuss the RT-QuIC alpha-synuclein technique, as well as biomarkers not directly related to the proteins aggregating in the diseases.

### 4.1. RT-QuIC Technique

RT-QuIC means “real-time quaking-induced conversion” and this technique, which was originally developed to detect the abnormal prion protein, is based on the “prion-like” effect of α-synuclein. It is important to know that from a pathophysiological point of view, physiological α-synuclein is transformed into “pathological” α-synuclein, i.e., it undergoes post-translational modifications, changes its conformation with the formation of beta-sheets, and aggregation phenomena leading to Lewy bodies formation. Moreover, the technique uses Thioflavin T which is a fluorophore that binds to these beta-sheets. Thus, the RT-QuIC technique consists of adding patient CSF to recombinant α-synuclein and Thioflavin T. In case of the presence of “pathological” α-synuclein in the sample, it will transform the recombinant α-synuclein into “pathological” α-synuclein. Thioflavin T binding increases as this transformation proceeds. The increase in fluorescence emitted following this binding will be read in real-time by a spectrometer. The transformation of recombinant α-synuclein into pathological α-synuclein can take several days, thus the reading is done under agitation at 42 °C for at least 5 days.

Thus, using this technique, research has shown a sensitivity of 92–93% and a specificity of 96–100% depending on the study between DLB and AD patients [56,57,58,59,60]. Thus the routine development of this technique could have a major impact on the diagnosis of synucleinopathies and DLB in particular.

### 4.2. Potentially Interesting Biomarkers in the Differential Diagnosis between AD and DLB

Some elements of the CSF, despite a lack of direct specificity with one or the other disease, are apparently able to differentiate them. Currently, some biomarkers may be of interest; however, for some of them, they were only found in a single study and further research is needed to confirm their discriminatory power.

#### 4.2.1. Alpha-Synuclein Protease

Neurosin is one of the few proteins on the list that still has a link to alpha-synuclein since it is an alpha-synuclein-degrading protease. Neurosin levels appear to be significantly lowered in synucleinopathies (DLB, PD, PD dementia) compared to AD patients and controls [20].

#### 4.2.2. Neuroinflammation

YKL-40, also called chitinase 3 like protein 1 (CHI3L1) or cartilage glycoprotein 39 (HC gp-39), is a secreted glycoprotein with glycosyl hydrolase functions. In the brain, YKL-40 is mainly expressed by astrocytes and plays a key role in inflammation, particularly in AD. In CSF, the YKL-40 level is significantly higher in AD patients compared to that observed in DLB patients or healthy controls [61,62].

Interleukin (IL)-6 has an important role in neuroinflammatory processes but also has a potential impact on cognitive function. Thus, it is not surprising that IL-6 secretion is impaired in neurodegenerative dementias involving neuroinflammation. In AD, activated glial cells induce elevated IL-6 expression, especially near senile plaques. Thus, IL-6 levels are significantly lower in the CSF of DLB patients than in AD patients and control subjects without dementia [63].

#### 4.2.3. Neurotransmitter Metabolites

Because DLB is characterized by the impairment of different neurotransmitter systems, it would seem interesting to study dopamine and its metabolites such as homovanilic acid (HVA) and 3,4-Dihydroxyphenylacetic acid (DOPAC), serotonin metabolites such as 5-hydroxyindolacetic acid (5-HIAA), epinephrine/norepinephrine metabolites such as 3-methoxy-4-hydroxyphenylethylene glycol (MHPG) and gamma-aminobutyric acid (GABA). Thus, all these elements have been shown to be significantly lower in DLB than in AD [64].

#### 4.2.4. Amino Acids and Neuropeptides

Cocaine and amphetamine-regulated transcript (CART) is a neuropeptide selectively expressed in hypothalamus neurons. These neurons project to regions thought to regulate mood, such as the prefrontal cortex, hippocampus and striatum. In humans, CART gene mutations are associated with depression and anxiety. It is interesting to note that in a recent MRI study, DLB patients had hypothalamic atrophy while this region was not affected in AD patients. This atrophy is associated with a significant reduction in CSF CART levels in DLB but not in AD [65].

Alterations in excitatory amino acid concentrations may be potentially involved in the pathogenesis of several neurodegenerative diseases, including AD, PD, amyotrophic lateral sclerosis, and MSA. Thus, patients with DLB have higher concentrations of asparagine (+25%) and glycine (+21%) in CSF compared to a control group [65]. It should be noted that this study did not include AD patients.

#### 4.2.5. Minerals and Metals

Several neurodegenerative diseases, including AD and PD, are characterized by altered homeostasis of certain metals in the brain and CSF. These changes appear to contribute, directly or indirectly, to increased oxidative stress, an important factor of neuronal toxicity. DLB patients have elevated levels of calcium, magnesium, and copper in the CSF compared to control and AD patients. The combination of calcium and magnesium concentrations in the CSF distinguishes DLB from AD with a sensitivity of 93% and a specificity of 85% [65].

#### 4.2.6. Synaptic Proteins

Chromogranin A (CgA) is a neuroactive glycoprotein present in neuroendocrine cells and in synaptic vesicles of neurons. CgA plays a role in the stabilization of secretory vesicles and the synaptic activity modulation. This protein seems to be significantly higher in DLB patients than in control, PD and PDD patients [66]. Note the absence of Alzheimer’s patients in this study, but it would seem that CgA is also increased because of the MCI stage of AD patients [67]. Further studies would therefore be necessary to determine whether CgA would discriminate between DLB and AD.

Neurogranin is a postsynaptic protein that binds to calmodulin and plays an important role in memory by regulating synaptic plasticity and learning. Thus, the neurogranin release into the CSF is a likely reflection of synaptic dysfunction or neuronal degeneration. Despite the presence of synaptic dysfunction in DLB, neurogranin levels in the CSF of AD patients are significantly higher than in those affected by DLB [61]; they are probably related to more extensive neuronal death processes during AD.

#### 4.2.7. Others

Visinin-like protein 1 (VILIP-1) belongs to the family of neuronal calcium sensors. The entry of calcium cations into the cell induces a reversible translocation of VILIP-1 to membrane components of the cell. In this way, VILIP-1 modulates the signaling cascade in neurons, resulting in the regulation of neuronal ion channels, neuronal growth, synaptic plasticity, and the activation of cyclic adenosine monophosphate (cAMP) as well as cyclic guanine monophosphate (cGMP) signaling pathways. One study showed that the concentration of VILIP-1 in CSF was significantly increased in AD patients compared to normal controls and DLB patients [68,69].

A recent proteomic study has identified some biomarkers that may be of interest in the differential diagnosis of DLB [70]. The authors identified VGF, SCG2, NPTX2, NPTXR, PDYN and PCSK1N as possible biomarker candidates, with the NPTX2, VGF, SCG2, PDYN panels being the most relevant for differentiating DLB from AD [70,71]. All of these biomarker candidates are likely to be decreased in the CSF of DLB patients compared to controls and AD groups. However, several of them are also decreased in the CSF of AD patients compared to control patients, suggesting an even greater decrease for DLB patients. NPTX2 consistently shows decreased CSF concentrations in AD [72,73], Boiten and colleagues showed no difference in CSF NPTX2 levels between DLB and AD [74]. VGF and SCG2 also showed a decrease in CSF in AD patients [75,76], so among the potential biomarkers, only PDYN seems to remain unchanged in AD (no publications have been able to show any variation in the CSF levels of AD patients). VGF and SCG2 (but also PCSK1N) are members of the chromogranin/secretogranin family and play a role in the regulated secretory pathway of peptides, hormones, neurotransmitters and growth factors. VGF is a neurosecretory protein. VGF and peptides derived from its processing play many roles in neurotransmitter release, energy homeostasis, and regulation of gastrointestinal function but also in neurogenesis and neuroplasticity associated with learning, memory, depression and chronic pain [77,78]. The second biomarker candidate is SCG2, which is a neuroendocrine protein of the granin family that regulates the biogenesis of secretory granules playing a role in inflammatory responses and in the regulation of blood pressure [77]. NPTX2 (and NPTXR) are members of the neuronal pentraxin family [79]. NPTX2 plays role in the modification of cellular properties that underlie long-term plasticity promoting the formation of new excitatory synapses (through the glutamate-gated channels), synaptic homeostatic plasticity and regulation of AMPA-type receptors clustering at established synapses [80]. The last biomarker candidate identified is PDYN. This protein is a member of the enkephalin family that competes with and mimics the effects of opioid drugs. Enkephalins play a role in a number of physiologic functions, including pain perception and responses to stress [81,82].

Table 3 summarizes the variations in the levels of these different biomarkers between AD and DLB compared to control subjects.

## 5. Conclusions

The diagnosis of DLB is primarily clinical. The use of imaging biomarkers or CSF is only useful in case of strong diagnostic doubt with another disease. The use of these biomarkers is not meant to rule out the diagnosis of DLB due to their imperfect specificity or sensitivity.

AD biomarkers are useful to distinguish AD from DLB, but specific DLB markers are sorely lacking. Numerous biomarkers are currently under investigation. To be considered specific, they must be able to distinguish DLB patients not only from those with AD but also from controls and other dementias. Among the “candidate” biomarkers, some are able to differentiate DLB from AD: they are therefore of interest in this precise differential diagnosis. But some have no capacity to discriminate DLB from normal aging because they lack real DLB specificity. These include biomarkers such as YKL-40, neurogranin, or VILIP-1. On the other hand, biomarkers such as neurosine, IL-6, CART, VGF, SCG2, and PDYN appear interesting because they differentiate DLB patients from both AD and control patients. In the same way, RT-QuIC seems promising for the diagnosis of synucleinopathies. Other biomarkers may potentially be of interest but still need to be proven on more complete cohorts such as chromogranin A, as well as some amino acids (asparagine, glycine) and neurotransmitter metabolites (HVA, 5-HIAA, MHPG). Overall, further analysis is still needed to confirm and strengthen these results before these biomarkers can be used with confidence in clinical routine.

The most used imaging exam is the brain MRI [83]: its main interest is to eliminate a possible cerebral lesion evoking a differential diagnosis of DLB (cerebral tumor, hemorrhagic sequel). The existence of an ischemic vascular lesion should not prevent the diagnosis of DLB because these can be frequent [84]. Focal atrophy will not clearly orient the diagnosis of DLB, even if hippocampal atrophy will generally be discrete or even absent, and the insular atrophy could be of interest in the context of MCI patients. Among the scintigraphy examinations, the FP-CIT SPECT seems to be the most powerful. However, it is less useful in the prodromal stage.

Our article did not address EEG, polysomnography, and blood biomarkers. This does not prevent them from being very useful for the clinician. Polysomnographic demonstration of REM sleep without atonia is a predictor of Lewy-related pathology and has been incorporated in the revised DLB consensus criteria [6]. EEG is becoming more accurate for the diagnosis of DLB [81], and neurofilament blood testing is likely to be involved in the diagnosis of DLB [85], especially in the differential diagnosis of psychiatric diseases.

Nevertheless, it is essential to discover new biomarkers capable of discriminating DLB from AD, FTLD and psychiatric diseases to increase the differential diagnosis, especially if these biomarkers are relevant at an early stage, as it is now well established that early diagnosis is important for the therapeutic management of these patients. As there are treatments for each symptom of DLB, an accurate and early diagnosis will allow to propose adapted symptomatic treatments for a better quality of life from the early stages. In the Alzheimer’s Disease Neuroimaging Initiative (ADNI) study, initial neuropathological data showed that 46% of patients initially diagnosed with AD had Lewy bodies at autopsy, suggesting co-morbidity of AD and DLB. It is therefore likely that patients in AD treatment trials do not have a purely AD pathology. This has serious implications for the development of new therapies for AD and DLB. Better diagnosis will allow patients to be directed to appropriate therapeutic trials. Finally, 80% of patients with DLB have a negative sensitivity to antipsychotic drugs (and for some patients to many other psychotropic drugs), even with treatments such as low-dose risperidone. Therefore, in the future, accurate and early diagnosis could reduce inappropriate neuroleptic prescribing errors and thus confusion, cognitive worsening, falls, swallowing problems and early death in patients with DLB.

## Figures and Tables

**Figure 1 ijms-23-06371-f001:**
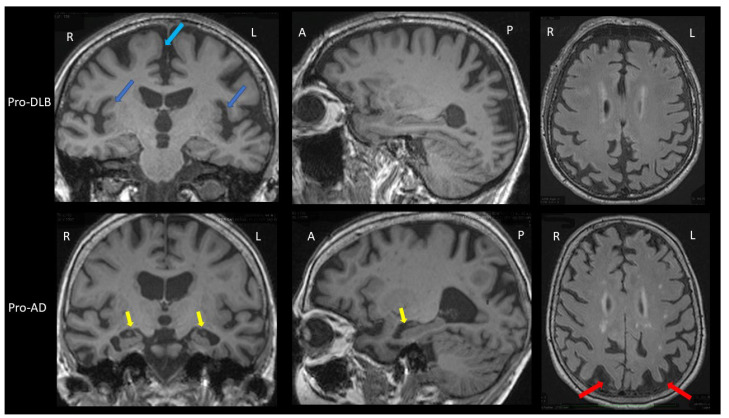
Brain MRI of patients with prodromal dementia with Lewy bodies and prodromal Alzheimer’s disease. **Upper part:** Brain MRI of a prodromal DLB (Pro-DLB) patient of 84 years with mild cognitive impairment (MMSE = 27/30), cognitive fluctuations, REM-sleep behavior disorder (RBD) and subtle Parkinsonism (UPDRS scale: rigidity 1/4 Froment’s maneuver, akinesia 1/4, attitude tremor). **Top left image:** T1 sequence, coronal section. The two dark blue arrows show the bilateral insular atrophy. Note the absence of hippocampal atrophy (Scheltens scale = 0/4) and the frontomesial atrophy (light blue arrow). **Middle image at top:** T1 sequence, sagittal section: absence of hippocampal atrophy and right parietal atrophy. **Top right image:** FLAIR sequence, axial section. There is no ischemic vascular lesion (Fazekas score = 0/3). Note the right parietal atrophy. **Lower part:** Brain MRI of a prodromal AD (Pro-AD) patient of 84 years old with mild cognitive impairment (MMSE = 29/30) with mainly verbal and visual memory storage disorders, and no Parkinsonism, no fluctuation, no RBD and no hallucination. **Bottom left image:** T1 sequence, coronal section. The two yellow arrows show the hippocampal atrophy (Scheltens scale = 3/4 for the left hippocampus and 2/4 for the right one). Note the subtle insular atrophy and frontomesial atrophy. **Bottom middle image:** T1 sequence, sagittal image. The yellow arrow shows the hippocampal atrophy. Note the parietal atrophy. **Bottom right image:** FLAIR sequence, axial section. The two red arrows show the bilateral parietal atrophy. There are few microvascular ischemic lesions as hypersignals (Fazekas score = 1/3). Comparison of the two patients shows greater hippocampal atrophy in the pro-AD patient, and greater insular atrophy in the pro-DLB patient.

**Figure 2 ijms-23-06371-f002:**
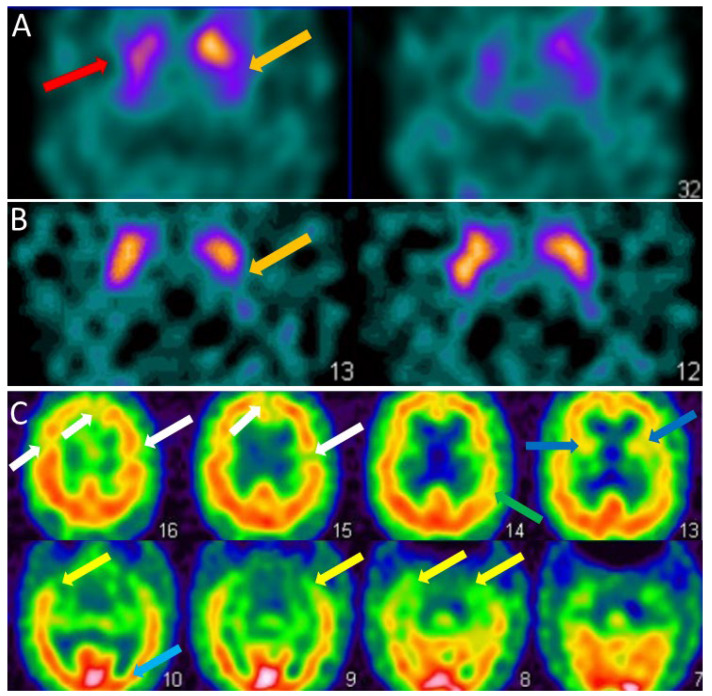
SPECT of patients with dementia with Lewy bodies. (**A**). Brain FP-CIT SPECT (DAT-scan) of a DLB patient of 81 years with mild dementia, cognitive fluctuations, and Capgras syndrome. Axial section FP-CIT SPECT shows lower right striatal, i.e., caudate and putamen, uptake (red arrow) and lower left putamen uptake (orange arrow). (**B**). Brain FP-CIT SPECT (DAT-scan) of a prodromal DLB patient of 74 years with mild cognitive impairment, cognitive fluctuations, and subtle Parkinsonism (UPDRS scale: rigidity 1/4, akinesia 1/4, no tremor). Axial section FP-CIT SPECT shows lower left putamen uptake (orange arrow). (**C**). Brain perfusion SPECT (neurolite) of the same prodromal DLB patient of 74 years with mild cognitive impairment, cognitive fluctuations, and subtle Parkinsonism (UPDRS scale: rigidity 1/4, akinesia 1/4, no tremor). Axial section perfusion SPECT showing a mity aspect with diffuse hypoperfusion involving frontal (white arrows), insular (dark blue arrows), parietal (green arrow), temporal (yellow arrow) and occipital lobes (subtle, light blue arrow).

**Table 1 ijms-23-06371-t001:** Diagnostic criteria of DLB.

Clinical Characteristics	
Essential Criterion	
Cognitive decline of sufficient severity to interfere with activities of daily living. The deficits frequently concern attentional, executive and visuospatial abilities.	
**Core Criteria**	**Supportive Criteria**
➢ Cognitive fluctuations with significant changes in attention and alertness ➢ Recurrent visual hallucinations that are typically well formed and detailed ➢ REM sleep behavior disorder, which may precede cognitive decline ➢ One or more characteristics of parkinsonian syndrome: bradykinesia, rest tremor, rigidity	➢ Severe sensitivity to antipsychotics ➢ Postural instability ➢ Repeated falls ➢ syncope or other transient episodes of unresponsiveness ➢ Severe autonomic dysfunction: constipation, orthostatic hypotension, urinary incontinence ➢ Hallucinations in other sensory modalities (ex: auditory hallucination) ➢ hypersomnia ➢ hyposmia ➢ Systematized delirium: delirium where the delusional ideas are organized, giving the impression of coherence ➢ Apathy, anxiety, depression
**Biomarkers**	
**Indicative**	**Supportive**
➢ Reduced dopamine transporter uptake in basal ganglia demonstrated by SPECT or PET ➢ Abnormal (low uptake) 123Iodine-MIBG myocardiac scintigraphy ➢ Polysomnographic confirmation of REM sleep without atonia	➢ MRI, CT-scan: relative preservation of medial temporal lobe structures (unlike AD patients) ➢ Generalized low uptake on SPECT/PET perfusion/metabolism scan with reduced occipital activity +/- the cingulate island sign on FDG-PET imaging ➢ EEG: Prominent posterior slow-wave activity with periodic fluctuations in the pre-α/θ range

Core criteria: the presence of two of the symptoms listed are essential for the probable diagnosis, and the presence of only one of them for the possible diagnosis of DLB. Suggestive criteria (green headband) can lead to a diagnosis of probable DLB (presence of at least one core criterion, and one of these suggestive criteria), or possible DLB (presence of at least one of these criteria without core criteria).EEG: Electroencephalography; FDG: Fluorodeoxyglucose (18F); MRI: Magnetic Resonance Imaging; PET scan: positron emission tomography; REM: rapid eye movement; SPECT: Single photon emission computed tomography; 123 Iodine-MIBG: 123 iodine-metaiodobenzylguanidine adapted from [6].

**Table 2 ijms-23-06371-t002:** Neuroimaging and diagnosis of dementia with Lewy bodies.

	Prodromal DLB	Validity	DLB dementia	Validity	References
**Brain MRI T1**	Insular atrophy	Not demonstrated	No or mild hippocampal atrophy	Sensitivity = 64% Specificity = 68% (compared to AD)	[30]
**Brain MRI SWI**	Loss of the swallow tail sign	Not demonstrated	Loss of the swallow tail sign	Sensitivity = 63% Specificity = 75% (compared to AD)	[32]
**FP-CIT SPECT (DAT-scan)**	Presynaptic striatal dopaminergic decrease	Sensitivity = 54.2% Specificity = 89.0% (compared to prodromal AD)	Presynaptic striatal dopaminergic decrease	Sensitivity = 77.7% Specificity = 90.4% (compared to AD)	[36,39]
**Perfusion SPECT**	Occipital hypoperfusion	Not demonstrated	Occipital hypoperfusion	Sensitivity = 74.0% Specificity = 82.0% (compared to AD)	[42]
**FDG-PET**	Occipital hypometabolism and Cingulate Island Sign	Not demonstrated	Occipital hypometabolism and Cingulate Island Sign	Sensitivity = 77.0% Specificity = 80.0% (compared to AD)	[47]
**Synuclein-PET**	Cortical and basal ganglia accumulation?	Not existing	Cortical and basal ganglia accumulation?	Not existing	
**MIBG scintigraphy**	Decrease cardiac sympathetic activity	Sensitivity = 46.2% Specificity = 88.0% (compared to prodromal AD)	Decrease cardiac sympathetic activity	Sensitivity = 68.9% Specificity = 87.0% (compared to AD)	[53,55]

**Table 3 ijms-23-06371-t003:** Summarizes the variations of potential biomarkers in AD and DLB CSF (- means no change, ↗ means increased compared to controls, ↘ means decreased compared to controls, ? means no data).

Biomarkers	AD	DLB	References
YKL-40	↗	-	[61,62]
neurogranin	↗	-	[61]
VILIP-1	↗	-	[68,69]
Magnesium, calcium, copper	-	↗	[65]
Neurosin	-	↘	[20]
Il-6	-	↘	[63]
CART	-	↘	[65]
Chromogranin A	↗	↗	[66,67]
Asparagine, glycine	?	↗	[65]
HVA, 5-HIAA et MHPG		↘ compared to AD	[64]
NPTX2, VGF, SCG2,	↘	↘(↘)	[70]
PDYN		↘	[70]
RT-QuIC	-	↗	[56,57,58,59,60]

## Data Availability

Not applicable.

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
