# Peer review of "Biomarkers of Dementia with Lewy Bodies: Differential Diagnostic with Alzheimer’s Disease"

_ijms, 2022, doi:10.3390/ijms23126371_

Round 1

Reviewer 1 Report

An update on current and possible future diagnostic tools for Lewy body disease, the second most prevalent neurodegenerative dementia, is always of interest. I thank the authors for their work and highlight the quality of the images incorporated into the manuscript. 

Below are my suggestions in an attempt to improve the quality of the manuscript: 

1- Providing information on the search strategy employed would be of interest. Although it is not a systematic review, it would help the reader to understand the criteria used to select the articles included in the review, even if only in supplementary material (I would recommend a combination of both a text and figure about search strategy).  
2- I think the initial introductory section would be improved with another structure: a) epidemiological data on DLB, b) diagnostic criteria for lewy body prodromal disease and dementia (can be in a panel and summarized); c) differential diagnosis with AD and PD (point out differences in neuropsychological assessment pattern in case of AD, differences in the level of temporal relationship of onset of motor symptoms in case of PD; other common symptoms in case of DLB such as: REM sleep behavior disorder, cognitive fluctuations and also sensations of presence and passage); d) diagnostic tools available at the present time and e) information on therapeutic management in a simple way (what is offered to these patients in clinical practice? What precautions are necessary?) 
3- In the section on neuroimaging biomarkers, a summary table on the findings that allow a differential diagnosis between dementia with Lewy bodies and Alzheimer's disease, and also indicating the timing of these changes (prodromal phases versus dementia phases in both cases) would be useful. Ordering the text following the same outline would also help the reader. Right now it gives the impression that it is a bit disorganized.
4- In the section on CSF biomarkers, I think it would be useful to discuss the relevance of the coexistence of AD pathology, which is not at all uncommon in patients with DLB. 
5- Section 4.2. could be restructured in different subsections such as biomarkers related to neuroinflammation (the usefulness of GFAP is not suggested, for some specific reason?), related to synaptic damage... it would be more understandable. 
6- In the table provided, it would be of great interest to point out not only the direction of the change but also the sensitivity and specificity of each of the postulated biomarkers (same for neuroimaging biomarkers). Try to improve the visualization of the table (somewhat confusing) and add information on the abbreviations used in the table in the table footer. 
7- For all biomarkers analyzed it would be interesting to know not only diagnostic utility in prodromal and dementia stage, but potential prognostic value given to them. 
8- A mention of PSG in the introduction when discussing diagnostic criteria would be reasonable. On the other hand, if mention is made of EEG, it should be pointed out what differences are expected to be found with respect to AD and other neurodegenerative dementias. 
9- Commenting on the usefulness of plasma biomarkers in a more extensive way is also necessary. There are multiple publications suggesting that NfL is nonspecific to discriminate between different neurodegenerative dementias but not in the case of p-tau181 or other isoforms of p-tau,... A brief mention of other potential biomarkers such as epigenetic biomarkers could also be of interest. 
10- A comment on the importance of early diagnosis would be interesting. Also linking advances in diagnosis with possible advances in treatment would also be of great interest.

Thus I conclude the first part of my review, but not before thanking the authors for their efforts in addressing this exciting topic. 

Author Response

Reviewer 1

An update on current and possible future diagnostic tools for Lewy body disease, the second most prevalent neurodegenerative dementia, is always of interest. I thank the authors for their work and highlight the quality of the images incorporated into the manuscript. 

Below are my suggestions in an attempt to improve the quality of the manuscript: 

  • Providing information on the search strategy employed would be of interest. Although it is not a systematic review, it would help the reader to understand the criteria used to select the articles included in the review, even if only in supplementary material (I would recommend a combination of both a text and figure about search strategy).  

In response to your request, we have added a methodology paragraph.

2- I think the initial introductory section would be improved with another structure: a) epidemiological data on DLB, b) diagnostic criteria for lewy body prodromal disease and dementia (can be in a panel and summarized); c) differential diagnosis with AD and PD (point out differences in neuropsychological assessment pattern in case of AD, differences in the level of temporal relationship of onset of motor symptoms in case of PD; other common symptoms in case of DLB such as: REM sleep behavior disorder, cognitive fluctuations and also sensations of presence and passage); d) diagnostic tools available at the present time and e) information on therapeutic management in a simple way (what is offered to these patients in clinical practice? What precautions are necessary?) 

The introduction has been improved and restructured according to your recommendations.

3- In the section on neuroimaging biomarkers, a summary table on the findings that allow a differential diagnosis between dementia with Lewy bodies and Alzheimer's disease, and also indicating the timing of these changes (prodromal phases versus dementia phases in both cases) would be useful. Ordering the text following the same outline would also help the reader. Right now it gives the impression that it is a bit disorganized.

A table summarising neuroimaging findings in the diagnosis of DLB has been added and the section on these biomarkers has been reorganised.

4- In the section on CSF biomarkers, I think it would be useful to discuss the relevance of the coexistence of AD pathology, which is not at all uncommon in patients with DLB. 
We agree with you, however the majority of the articles cited do not do post-mortem studies to confirm diagnoses and do not present any group with AD/DLB comorbidity. It seems difficult to us to discuss this issue without being able to rely on results.

5- Section 4.2. could be restructured in different subsections such as biomarkers related to neuroinflammation (the usefulness of GFAP is not suggested, for some specific reason?), related to synaptic damage... it would be more understandable. 
As you suggested, we have restructured paragraph 4.2. The GFAP protein is not discussed, because as we have indicated, we are highlighting molecules capable of discriminating AD from DLB, which is, to our knowledge, not the case for GFAP.

6- In the table provided, it would be of great interest to point out not only the direction of the change but also the sensitivity and specificity of each of the postulated biomarkers (same for neuroimaging biomarkers). Try to improve the visualization of the table (somewhat confusing) and add information on the abbreviations used in the table in the table footer. 

As sensitivity and specificity are not always given in the articles, unfortunately, we could not complete the table with these data.

7- For all biomarkers analyzed it would be interesting to know not only diagnostic utility in prodromal and dementia stage, but potential prognostic value given to them. 

All studies concerning CSF biomarkers have included DLB patients at the demented stage, and none of the cited papers have looked at the prognostic aspect of these biomarkers, so we cannot comply with your request.
8- A mention of PSG in the introduction when discussing diagnostic criteria would be reasonable. On the other hand, if mention is made of EEG, it should be pointed out what differences are expected to be found with respect to AD and other neurodegenerative dementias.

As agreed, a paragraph was added in the introduction on PSG and EEG.

9- Commenting on the usefulness of plasma biomarkers in a more extensive way is also necessary. There are multiple publications suggesting that NfL is nonspecific to discriminate between different neurodegenerative dementias but not in the case of p-tau181 or other isoforms of p-tau,... A brief mention of other potential biomarkers such as epigenetic biomarkers could also be of interest. 
The idea of the article is to highlight the biomarkers capable of discriminating AD from DLB (in this respect we have modified the title to make it clearer), at present blood biomarkers such as NFL do not make it possible to discriminate these two pathologies. Same problem with epigenetic markers with, moreover, studies, for most of them, done post-mortem on brain samples.

10- A comment on the importance of early diagnosis would be interesting. Also linking advances in diagnosis with possible advances in treatment would also be of great interest.

Indeed this point is very interesting, we have added a paragraph at the end of the discussion to address these issues.

Reviewer 2 Report

The review “Biomarkers of Dementia with Lewy Bodies” by Olivier Bousiges and Frédéric Blanc evaluates the importance of certain biomarkers regarding how informative they are to differentially diagnose DLB and AD. The authors focus mainly on summarizing brain imaging methods and CSF-based approaches. Overall, the authors give a comprehensive overview of the state-of-the-art methods which are used and discuss advantages and limitations. The manuscript is timely and useful for clinicians and scientists who work in related fields. However, usage of words and frequent typos make it challenging to read the paper. I would recommend thorough editing by a native English speaker.

Comments

  1. Since the methods were usually compared on their level of sensitivity and specificity, it would be very helpful for the (non-expert) reader to define the meaning of these words in the context of the discussed tests. A brief explanation in the introduction would be very helpful.
  2. The sentence “In addition, insular atrophy seems to be an interesting marker of prodromal DLB when comparing to prodromal AD and healthy elderly controls (Figure 1A)” refers to figure 1. However, figure 1 depicts just the brains of a prodromal DLB patient and DLB patient but not of controls or AD patients. It would be good to add additional images if possible or re-phrase the sentence.

Minor comments

  1. Generally, all abbreviations should be spelled out once and later used as abbreviations. In many cases, the abbreviations were never spelled out. Also, even when defined earlier in the manuscript, the abbreviations were not used but the words were spelled out. Examples: MCI is never spell out; Line 38: Parkinson's disease dementia (PDD), line 54: Parkinson’s Disease dementia OR: line 154: cingulate island sign (CIS), line 164 cingulate island sign (CIS).
  2. “Tableau II” should be replaced by “Table 1”. Also, it would be good to explain what “–“ means in comparison to “ “. No change? No data? Please add explanation in the text above the table.
  3. Line 19: The sentence “(...) relevance of AD biomarkers but also alpha-synuclein assay in DLB, (...)” should read “(...) relevance of AD biomarkers but also alpha-synuclein assays in DLB, (...)”
  4. Line 34: The sentence “(...) DLB criteria diagnostic have evolved over time (...)” should read “(...) DLB diagnostics criteria have evolved over time (...)”
  5. Line 71: The sentence “(...) be quite good (but at a late stage) for differentiating (...)” should read “(...) be quite good (at a late stage) for differentiating (...)”
  6. Line 87: the sentence “Studies show that α-synuclein may actually be relevant in discriminating AD from DLB, but the changes do not come from DLB patients, but from AD patients” is unclear. What does “but the changes do not come from DLB patients, but from AD patients” refer to?
  7. Line 94: the sentence “Thus, this review will look at all biomarkers other than Alzheimer's biomarkers or the total α-syn assay or the α-syn posttranslational modifications assay.” is unclear. What does “other than Alzheimer's biomarkers” mean?
  8. Line 120: The sentence “(...) , the interest of hippocampal atrophy decreases: sensitivity (...)” would be easier to understand: “(...) , hippocampal atrophy is less informative: sensitivity (...)”
  9. Line 129: The sentence “(...) can also appeared in DLB (...)” should read: “(...) can also appear in DLB (...)”
  10. Line 154: the sentence “Using image processing with software SPECT Z-score maps with a focus on the cingulate island sign (CIS, see infra), with a sensitivity and specificity of 92.3, and 76.9% (34).” is incomplete.
  11. Line 155: the sentence “But these results were lower in a second more powerful study with sensitivity and specificity of sensitivity and specificity of 50 and 73% (35).” is not understandable.
  12. Line 185: The sentence “(...) the sensitivity was 77,7% and the specificity of 90,4% (...)” should read: “(...) the sensitivity was 77,7% and the specificity was 90,4% (...)”
  13. Line 185: The sentence “(...) sensitivity was of 54,2% for probable and possible prodromal DLB, and specificity of 89% (...)” should read: “(...) sensitivity was 54,2% for probable and possible prodromal DLB, and specificity was 89% (...)”
  14. Line 199: The sentence “(...) at the whole striatum, caudate nucleus in front and putamen behind (...)” is unclear. Please correct.
  15. Line 185: The sentence “(...) , the sensitivity is: from 68,9% (...)” should read: “(...) the sensitivity ranges from 68,9% (...)”
  16. Line 223: I suggest to change “Although MIBG scintigraphy seems to be of interest, “ to “Although MIBG scintigraphy seems to be informative,”
  17. Line 226: “AS” should be corrected to “As”
  18. Line 233: The sentence “(...) There is many efforts to (...)” should read: “(...) There are many efforts to (...)”
  19. Line 252: The sentence “(...) the RT-QuIC technique consists in adding (...)” should read: “(...) the RT-QuIC technique consists of adding (...)”
  20. Line 258: The sentence “(...) agitation at 42°C during at least 5 days (...)” should read: “(...) agitation at 42°C for at least 5 days (...)”
  21. Line 276: The sentence “(...) but it would seem that CgA is also increased since the MCI stage of AD patients (58). (...)” should read: “(...) but it would seem that CgA is also increased because of the MCI stage of AD patients, in which CgA has been reported to be elevated (58). (...)”
  22. Line 333: The sentence “(...) also showed a decrease in CSF AD patients (...)” should read: “(...) also showed a decrease in CSF of AD patients (...)”
  23. Line 343: The sentence “(...) responses and in a regulation of blood pressure (...)” should read: “(...) responses and in the regulation of blood pressure (...)”
  24. Line 344: The sentence “(...) NPTX2 play role in the (...)” should read: “(...) NPTX2 plays a role in the (...)”

Author Response

Reviewer 2

The review “Biomarkers of Dementia with Lewy Bodies” by Olivier Bousiges and Frédéric Blanc evaluates the importance of certain biomarkers regarding how informative they are to differentially diagnose DLB and AD. The authors focus mainly on summarizing brain imaging methods and CSF-based approaches. Overall, the authors give a comprehensive overview of the state-of-the-art methods which are used and discuss advantages and limitations. The manuscript is timely and useful for clinicians and scientists who work in related fields. However, usage of words and frequent typos make it challenging to read the paper. I would recommend thorough editing by a native English speaker.

Comments

  1. Since the methods were usually compared on their level of sensitivity and specificity, it would be very helpful for the (non-expert) reader to define the meaning of these words in the context of the discussed tests. A brief explanation in the introduction would be very helpful.

We have added a methodology paragraph in which we have added some sentences to explain these terms.

  1. The sentence “In addition, insular atrophy seems to be an interesting marker of prodromal DLB when comparing to prodromal AD and healthy elderly controls (Figure 1A)” refers to figure 1. However, figure 1 depicts just the brains of a prodromal DLB patient and DLB patient but not of controls or AD patients. It would be good to add additional images if possible or re-phrase the sentence.

You are quite right, figure 1 has been redesigned and now shows the MRI of a prodromal DLB patient and a prodromal AD patient.

Minor comments

  1. Generally, all abbreviations should be spelled out once and later used as abbreviations. In many cases, the abbreviations were never spelled out. Also, even when defined earlier in the manuscript, the abbreviations were not used but the words were spelled out. Examples: MCI is never spell out; Line 38: Parkinson's disease dementia (PDD), line 54: Parkinson’s Disease dementia OR: line 154: cingulate island sign (CIS), line 164 cingulate island sign (CIS).
  2. “Tableau II” should be replaced by “Table 1”. Also, it would be good to explain what “–“ means in comparison to “ “. No change? No data? Please add explanation in the text above the table.
  3. Line 19: The sentence “(...) relevance of AD biomarkers but also alpha-synuclein assay in DLB, (...)” should read “(...) relevance of AD biomarkers but also alpha-synuclein assays in DLB, (...)”
  4. Line 34: The sentence “(...) DLB criteria diagnostic have evolved over time (...)” should read “(...) DLB diagnostics criteria have evolved over time (...)”
  5. Line 71: The sentence “(...) be quite good (but at a late stage) for differentiating (...)” should read “(...) be quite good (at a late stage) for differentiating (...)”
  6. Line 87: the sentence “Studies show that α-synuclein may actually be relevant in discriminating AD from DLB, but the changes do not come from DLB patients, but from AD patients” is unclear. What does “but the changes do not come from DLB patients, but from AD patients” refer to?
  7. Line 94: the sentence “Thus, this review will look at all biomarkers other than Alzheimer's biomarkers or the total α-syn assay or the α-syn posttranslational modifications assay.” is unclear. What does “other than Alzheimer's biomarkers” mean?
  8. Line 120: The sentence “(...) , the interest of hippocampal atrophy decreases: sensitivity (...)” would be easier to understand: “(...) , hippocampal atrophy is less informative: sensitivity (...)”
  9. Line 129: The sentence “(...) can also appeared in DLB (...)” should read: “(...) can also appear in DLB (...)”
  10. Line 154: the sentence “Using image processing with software SPECT Z-score maps with a focus on the cingulate island sign (CIS, see infra), with a sensitivity and specificity of 92.3, and 76.9% (34).” is incomplete.
  11. Line 155: the sentence “But these results were lower in a second more powerful study with sensitivity and specificity of sensitivity and specificity of 50 and 73% (35).” is not understandable.
  12. Line 185: The sentence “(...) the sensitivity was 77,7% and the specificity of 90,4% (...)” should read: “(...) the sensitivity was 77,7% and the specificity was 90,4% (...)”
  13. Line 185: The sentence “(...) sensitivity was of 54,2% for probable and possible prodromal DLB, and specificity of 89% (...)” should read: “(...) sensitivity was 54,2% for probable and possible prodromal DLB, and specificity was 89% (...)”
  14. Line 199: The sentence “(...) at the whole striatum, caudate nucleus in front and putamen behind (...)” is unclear. Please correct.
  15. Line 185: The sentence “(...) , the sensitivity is: from 68,9% (...)” should read: “(...) the sensitivity ranges from 68,9% (...)”
  16. Line 223: I suggest to change “Although MIBG scintigraphy seems to be of interest, “ to “Although MIBG scintigraphy seems to be informative,”
  17. Line 226: “AS” should be corrected to “As”
  18. Line 233: The sentence “(...) There is many efforts to (...)” should read: “(...) There are many efforts to (...)”
  19. Line 252: The sentence “(...) the RT-QuIC technique consists in adding (...)” should read: “(...) the RT-QuIC technique consists of adding (...)”
  20. Line 258: The sentence “(...) agitation at 42°C during at least 5 days (...)” should read: “(...) agitation at 42°C for at least 5 days (...)”
  21. Line 276: The sentence “(...) but it would seem that CgA is also increased since the MCI stage of AD patients (58). (...)” should read: “(...) but it would seem that CgA is also increased because of the MCI stage of AD patients, in which CgA has been reported to be elevated (58). (...)”
  22. Line 333: The sentence “(...) also showed a decrease in CSF AD patients (...)” should read: “(...) also showed a decrease in CSF of AD patients (...)”
  23. Line 343: The sentence “(...) responses and in a regulation of blood pressure (...)” should read: “(...) responses and in the regulation of blood pressure (...)”
  24. Line 344: The sentence “(...) NPTX2 play role in the (...)” should read: “(...) NPTX2 plays a role in the (...)”

Thank you very much for all these corrections which we have taken into account to modify the manuscript.

Round 2

Reviewer 1 Report

I think that the authors have incorporated most of the suggestions of the reviewers. Thanks to them I do believe that the quality of the manuscript has improved significantly. I don't have nothing more to add. 

This manuscript is a resubmission of an earlier submission. The following is a list of the peer review reports and author responses from that submission.